# The experience of loneliness among people with psychosis: Qualitative meta-synthesis

**Cheuk Ying Lee**[1]*, **Nafiso Ahmed**[1], **Sarah Ikhtabi**[1], **Phoebe Duffett**[1], **Yazan Alhilow**[1], **Leonie Richardson**[1], **Alexandra Pitman**[1,2], **Brynmor Lloyd-Evans**[1], **Sonia Johnson**[1,2]

1 Division of Psychiatry, University College London, London, United Kingdom, 2 North London NHS Foundation Trust, London, United Kingdom

* nafiso.ahmed@ucl.ac.uk

**Data Availability Statement:** All relevant data are within the manuscript and its Supporting Information files.

## Abstract

### Background

Loneliness can be a significant challenge for people living with psychosis. We currently lack evidence about how to help with this. Understanding the origins, experience, and impact of loneliness in this group is helpful in developing strategies to reduce it. We therefore aimed to conduct a meta-synthesis of the qualitative literature exploring the nature of loneliness, and the factors contributing to the development and maintenance of loneliness, among people living with psychosis.

### Methods

We searched four electronic databases (MEDLINE, Embase, PsycINFO and Web of Science) for studies meeting pre-defined eligibility criteria. We adhered to the Preferred Reporting Items for Systematic reviews and Meta-Analysis (PRISMA) guidelines, and appraised the quality of each eligible study using the Critical Appraisal Skills Programme (CASP) checklist. We conducted thematic synthesis to combine the included qualitative findings to generate key overarching themes.

### Results

We included 41 articles and our analysis generated six meta-themes: (1) loneliness in the form of disconnection, (2) a rejecting and stigmatising external world, (3) loneliness and acute psychotic episodes and symptoms, (4) thwarted longing for connection, (5) paradoxes of loneliness and psychosis, and (6) well-being embedded in the social world. Our findings suggest a vicious cycle in which loneliness, psychosis and social withdrawal can be mutually reinforcing. Reciprocity of and continuity in relationships were valued by those with psychosis, while self-stigma and experiences of rejection appeared to exacerbate loneliness and impede recovery.

### Conclusion

Our meta-synthesis offers insights into how people living with psychosis are affected by loneliness. Tailored interventions are needed, which should focus on supporting people in

**Funding:** SJ, AP, BLE and NA receive salary support from the Loneliness & Social Isolation in Mental Health Research Network, which was funded by UK Research and Innovation (Grant reference: ES/S004440/1) for the period during which this review was commenced and is now funded by the National Institute for Health Research (NIHR) University College London Hospitals (UCLH) Biomedical Research Centre (BRC). SI is supported by the Economic and Research Social Council (ESRC) UKRI (Grant reference: ES/P000592/1) and UCL. The funders had no role in study design, data collection and analysis, decision to publish, or preparation of the manuscript.

**Competing interests:** The authors have declared that no competing interests exist.

overcoming internal barriers to connection and low self-esteem, challenging stigma and self-stigma, and linking people to social support according to needs and preferences.

## Introduction

Loneliness is a subjective emotional state that arises when there is a discrepancy between desired and achieved patterns of social interaction [1, 2]. Loneliness is distinct from objective concepts related to social connectedness, such as social network size or frequency of social contact [3]. According to the social neuroscientist Cacioppo and his colleagues [4], intimate, relational and collective dimensions of loneliness can be distinguished, corresponding to the quality of intimate relationships, friendships and family connections, and belongingness in wider society.

Although distressing, transient loneliness can be adaptive and serves as a motivator to obtain and maintain social relationships [5]. However, loneliness is persistent for some, and is associated with a range of physical and mental health difficulties [4], with established longitudinal relationships with all-cause mortality and cancer mortality [6], depression [7], suicidality [8], and associations with several other mental health conditions [9].

Most of the current research on loneliness and mental health focuses on links between loneliness and depression [7]. However, there is growing evidence that loneliness rates are also high among individuals with psychosis [10]. Psychosis is an umbrella term for conditions (including schizophrenia) characterised by a range of symptoms categorised as positive (e.g., hallucinations, delusions, disorganised behaviour) or negative (e.g., social anhedonia, withdrawal, blunted affect) [11]. Two systematic reviews and meta-analyses of cross-sectional studies aiming to investigate the relationships between loneliness and related psychological factors in psychosis have found significant positive associations between loneliness and the severity of psychotic symptoms, low self-esteem, low self-efficacy, and high internalized stigma [12, 13]. Studies have also found that individuals with psychosis face difficulty developing and sustaining rewarding relationships, have restricted access to social support outside of mental health services [14, 15], and experience worse quality of life because of loneliness [16]. As with people with common mental disorders and a diagnosis of a 'personality disorder', people with psychosis persistently report experiencing loneliness and identify this as a major challenge in recovery [17–19]. Together, this evidence suggests that interventions focused on reducing loneliness could be beneficial to the recovery and well-being of people with psychosis [16].

Qualitative research investigating the views of mental health practitioners in early intervention services has suggested that loneliness is not routinely discussed in early intervention services for first-episode psychosis, and there is a lack of targeted approach for reducing loneliness [20]. Potential psychological and psychosocial approaches to alleviating loneliness among people with mental health problems include supporting people to change negative social cognitions, social skills training and psychoeducation, having socially-focused supporters to enhance social support, and wider community approaches to facilitate social participation [21]. Thus far, there is limited evidence about which interventions are effective, and in which mental health conditions [22]. The quantitative evidence on the relationship between loneliness and psychosis identifies a complex range of psychosis-specific influences, including positive and negative symptoms, maladaptive social cognition, hampered sense of pleasure, and the impacts of stigma and self-stigma [10, 23, 24]. Studies have also suggested a self-reinforcing cycle in which loneliness increases the risk of psychosis onset, aggravates symptoms,

and maintains the severity of psychosis [17, 25]. The complexity of relationships between loneliness and psychosis suggests that approaches to addressing this need to draw on an understanding of the distinctive features of experiences of loneliness among people with psychosis, and the ways in which these experiences may vary by the nature of symptoms and social context.

Qualitative studies are a means to understand the relationship between loneliness and mental health, and how this varies between individuals [26]. Such studies also yield evidence on what kinds of interventions people may perceive as acceptable and helpful. We therefore aimed to provide a synthesis of such evidence in order to inform further work to understand the relationship between loneliness and psychosis, as well as the potential directions for intervention development. We aimed to answer the following two research questions: How do individuals with psychosis experience loneliness? What factors do people with psychosis perceive as contributing to the development and persistence of loneliness?

## Methods

We adhered to the Preferred Reporting Items for Systematic reviews and Meta-Analysis (PRISMA) guidelines [27], and pre-registered our protocol on PROSPERO (CRD42022335801). The completed PRISMA checklist is available in S1 Appendix. The reporting of this meta-synthesis adhered to the ENTREQ guideline [28] (see S2 Appendix).

### Design

We conducted a meta-synthesis of qualitative research investigating the experience of loneliness among people with psychosis. Qualitative studies enable researchers to acquire first-hand insights into the world experienced by research participants [29]. Meta-synthesis is a rigorous research method developed for analysing and synthesizing results from existing qualitative research, and to construct higher-order meaning through identifying overarching themes [30–33]. It is adapted from thematic synthesis of primary data and entails organizing free codes of related areas into descriptive themes, then further interpreting patterns to generate analytical themes [34]. It is an iterative, cyclical process in which data across studies are compared and contrasted to achieve emerging understanding that can explain all initial descriptive themes [31].

In this study, qualitative data derived from eligible published studies were synthesised using an established six-step process for meta-synthesis, aiming for an appropriate balance between objectivity and researchers' subjectivity [31]. The steps are: (1) formulating research question and selection criteria, (2) selecting research based on selection criteria, (3) assessing quality of research, (4) extracting and presenting formal data, (5) analysing data, and (6) writing the synthesis [30].

### Database and search strategy

We searched four electronic databases (MEDLINE, Embase, PsycINFO and Web of Science) from inception to 18 May 2022, and updated the search on 10 November 2023. Search terms were established after a preliminary test search, which enabled the team to review concepts in pertinent research to encompass all relevant terms. Team composition is described under Reflexivity below. Search terms captured three key concepts: (i) psychosis, (ii) loneliness and (iii) qualitative research. To achieve comprehensive retrieval of relevant papers, we included conceptually overlapping or closely-related constructs such as social support and social isolation, but only extracted data specifically focused on the experience of loneliness for inclusion in the study. S3 Appendix displays search terms in full, including free text and MeSH terms.

**Table 1. Inclusion and exclusion criteria.**

| | Inclusion | Exclusion |
|---|---|---|
| **Study type** | • Any research studies with a qualitative research design to explore a participant's experience of loneliness, including any data collection methods (e.g., interviews and focus groups) and any methods of qualitative analysis of data (e.g., thematic analysis and interpretive phenomenological analysis)<br>• Qualitative data from mixed methods research | • Existing qualitative meta-syntheses or reviews<br>• Case studies or ethnographic exploration with only one participant<br>• Conference abstracts, PhD theses, dissertations or other types of grey literature<br>• Studies evaluating an intervention without data on loneliness prior to participation or reflection on past experience of loneliness<br>• Papers solely using a quantitative method |
| **Participants** | • At least 50% study participants had a clinical diagnosis of a psychotic disorder or meeting threshold criteria on an established diagnostic screening tool or symptom severity measure, or self-reported diagnosis or reported that they used services for people with psychosis<br>• Individuals diagnosed with schizophrenia, schizoaffective disorder, schizotypal personality disorder, bipolar disorder with psychotic features, depressive psychosis, delusional disorders, and other nonorganic psychosis, including both long-term, established psychosis and first-episode psychosis<br>• No age, gender or ethnicity restrictions | • Studies including qualitative data collected only from health care professionals and partners or other individuals with close experience of interaction with people with psychosis |
| **Concept** | • At least part of the results section concerned participants' current or retrospective experience of loneliness or closely related themes, including perceived social isolation and social support | • Papers solely focusing on the objective presence or absence of social support, as well as the non-psychological predictors or causes of loneliness<br>• Papers presenting data of loneliness fleetingly or not at all. For instance, a study with one participant saying; "I feel lonely because I'm psychotic" would not be deemed as being detailed enough to convey anything meaningful about the experience of loneliness |
| **Context** | • Studies in any geographical or cultural settings | |
| **Language** | • English only | • Non-English |

## Selection: Inclusion and exclusion criteria

Titles and abstracts of all identified studies were screened against inclusion and exclusioncriteria (see Table 1), followed by full-text screening.

## Data screening

After searching the four databases, we imported search results using a systematic review software package, EPPI-Reviewer [35], where references were deduplicated. Titles and abstracts of all citations were screened by the first author (CYL) against inclusion criteria. A second author (SI) independently screened a randomly-selected 10% of titles and abstracts to check for agreement. Then, full text of all potentially eligible papers were retrieved and screened independently in duplicate by CYL, and one of five reviewers (LR, NA, SI, PD,YA). Inter-rater agreement was assessed during the processes of screening, extraction and quality appraisal. Any disagreements were resolved through discussion until consensus reached.

## Extracting key features of eligible papers

We developed a data extraction proforma to summarize information and characteristics of each eligible paper, including citation, study setting and aim(s), sample size and type of diagnosis, sample characteristics, data collection and analysis methods, main theme(s), and quality assessment. These features of each eligible study were identified by CYL and checked for accuracy by another member of the team. Any discrepancies were resolved through discussion until consensus was reached.

## Quality appraisal

We conducted a quality appraisal of each eligible study using the Critical Appraisal Skills Programme (CASP) checklist; a widely used tool for rapidly appraising the quality of all types of

qualitative studies [36]. Studies were assessed on 10 items categorised as: validity, results, and value of the research. For each item, a score of 2 denotes the criterion is fully met, a score of 1 denotes the criterion is partially met, and a score of 0 denotes the criterion is not met. The higher the CASP score, the greater the quality rating of the study. We did not exclude studies based on low quality as this review aimed for a comprehensive synthesis of relevant experiences reported in all eligible papers. However, the CASP checklist was used to improve rigor of our meta-synthesis through presenting findings in the context of quality assessment of each study, as suggested in methodological guidance [37, 38]. Two reviewers conducted quality assessment of each included study independently, and discrepancies were resolved through discussion until consensus reached.

## Data extraction and analysis

One researcher (CYL) familiarised themselves with all qualitative data by carefully reading each included article, then identified and extracted findings relevant to the study research questions (direct quotations and authors' interpretations) of all eligible papers into a qualitative data analysis software package, NVivo 14 [39]. A second researcher (SI) independently assessed a randomly-selected 10% of eligible studies to establish consistency in decisions about which passages to import into the NVivo dataset, resolving any discrepancies through discussion.

Data synthesis followed a largely inductive approach that was guided by the two research questions. We did not seek to identify themes, codes or categories before analysis commenced, but instead to code data to generate themes capturing experiences of loneliness and any precipitating or perpetuating factors. Thematic synthesis was employed to analyse the relevant findings from the studies included in this review [34]. This approach was selected due to its flexibility in synthesising findings from diverse qualitative studies [31, 34] including the subjective and multifaceted experiences of loneliness, which can vary across individuals and contexts. Unlike framework synthesis, thematic synthesis allows themes to be developed inductively and without the constraints of a pre-existing framework, thus enabling the richness and depth of participants' experiences to be captured [34]. Similarly, meta-ethnography is often more focused on translating and synthesising studies into new theoretical models, which is less suited to the aim of this review.

Following the approach of thematic synthesis, CYL coded each line of text in accordance with its content and meaning, organised free codes into related categories to establish descriptive themes, with reference to the original named themes from the included papers; then developed new, overarching analytical themes [34]. Both first-order (direct quotations) and second-order (authors' interpretation) data were coded and assigned to developing analytical themes. SI independently coded data from a randomly-allocated 10% of the studies. The two researchers compared codes to create an initial coding framework, which was then refined through an iterative process to develop overarching analytical themes. Descriptive and analytical themes were identified and subsequently discussed with the wider review team, including meetings to address reflexivity.

## Reflexivity

The multidisciplinary nature of our research team minimized dominance of one perspective in decision-making throughout our methods or in interpretation of findings. The team consisted of academics and clinicians with a variety of perspectives on loneliness and psychosis based on their clinical and research experiences. CYL was a Hong Kong female MSc student in Clinical Mental Health Sciences interested in the topic of loneliness and psychosis. She is now a

practitioner working in primary mental health care. SJ and AP are white female psychiatrists with clinical and research experience relevant to the research topic: SJ's clinical practice is mainly with young adults with psychosis. NA is a black female postdoctoral researcher with clinical experience supporting individuals living with psychosis. SI is an Arab female postdoctoral researcher with an interest in the association between loneliness and mental health. BLE is a white male academic in the Division of Psychiatry from a mental health social work background. PD is a white female MSc student studying Clinical Mental Health Sciences, and LR is a white female who is a graduate of the same course. YA is a male Arab medical student with a strong interest in psychiatry and mental health, especially focusing on the relation between loneliness and negative mental health outcomes. Our team members do not have lived experience of psychosis, but a draft manuscript was read by two members of the Loneliness and Social Isolation in Mental Health Co-production Group (CoG), who wrote a lived experience commentary for this study (see discussion). Collaborative working helped generate a richer understanding of data, subjected analytical process to group reflexivity, and reduced the risk of personal bias and subjectivity inherent to meta-synthesis [31, 40].

# Results

## Study characteristics

Our initial database searches identified 10571 records, reduced to 7007 following de-duplication. After title and abstract screening, 484 papers were judged potentially relevant. Following full text screening, we identified 37 eligible studies. Our updated search on the 10 November 2023 identified four further eligible studies. Therefore, a total of 41 studies were included in the review. Inter-rater agreement between reviewers at full-text screening was high (90%), and through discussion 100% agreement was achieved. The list of studies excluded at full text screening is available in S4 Appendix. Fig 1 presents a PRISMA flow chart outlining the study selection process [27].

Study characteristics and detailed quality appraisal results are described in S5 and S6 Appendices, respectively. CASP assessment showed quality scores ranging from 9 to 20, with a median score of 17. Most studies stated aims and findings clearly, and employed appropriate qualitative research design. However, many received low scores for reflexivity.

The total number of participants in included studies was 701 (sample sizes between 5 and 90). Dates of publication ranged from 1995 to 2023. Studies originated from the United Kingdom (n = 10), the United States (n = 7), Canada (n = 5), Sweden (n = 4), Korea (n = 3), Australia (n = 2), Norway (n = 2), Taiwan (n = 2), China (n = 1), Denmark (n = 1), Israel (n = 1), the Netherlands (n = 1) and Poland (n = 1). One study analysed five transcripts from each of the following 15 sites: Brazil, Bulgaria, Cyprus, England, Finland, France, Greece, Italy, Lithuania, Malaysia, Romania, Slovakia, Slovenia, Turkey and the United States [41]. The majority of the studies (n = 39) involved individuals with either a diagnosis of psychosis or experience of psychosis that met the criteria for acceptance into early intervention services, whilst two studies included individuals with self-identified psychosis [42, 43]. One study exclusively explored the experiences of first-episode psychosis among black men [44], and one study investigated the experiences of individuals with schizophrenia coping with COVID-19 [45]. Four studies explicitly aimed to explore loneliness among individuals with psychosis [46–49]. The remainder had a broader aim to explore participants' conceptualization of interpersonal relationships as people living with psychosis, subjective experience of stigma attached to psychosis, or general experience of living with psychosis. Most studies (n = 39) collected data using semi-structured or unstructured interviews. One study was a secondary analysis of existing qualitative data from semi-structured interviews [50]. A variety of qualitative analytic approaches was

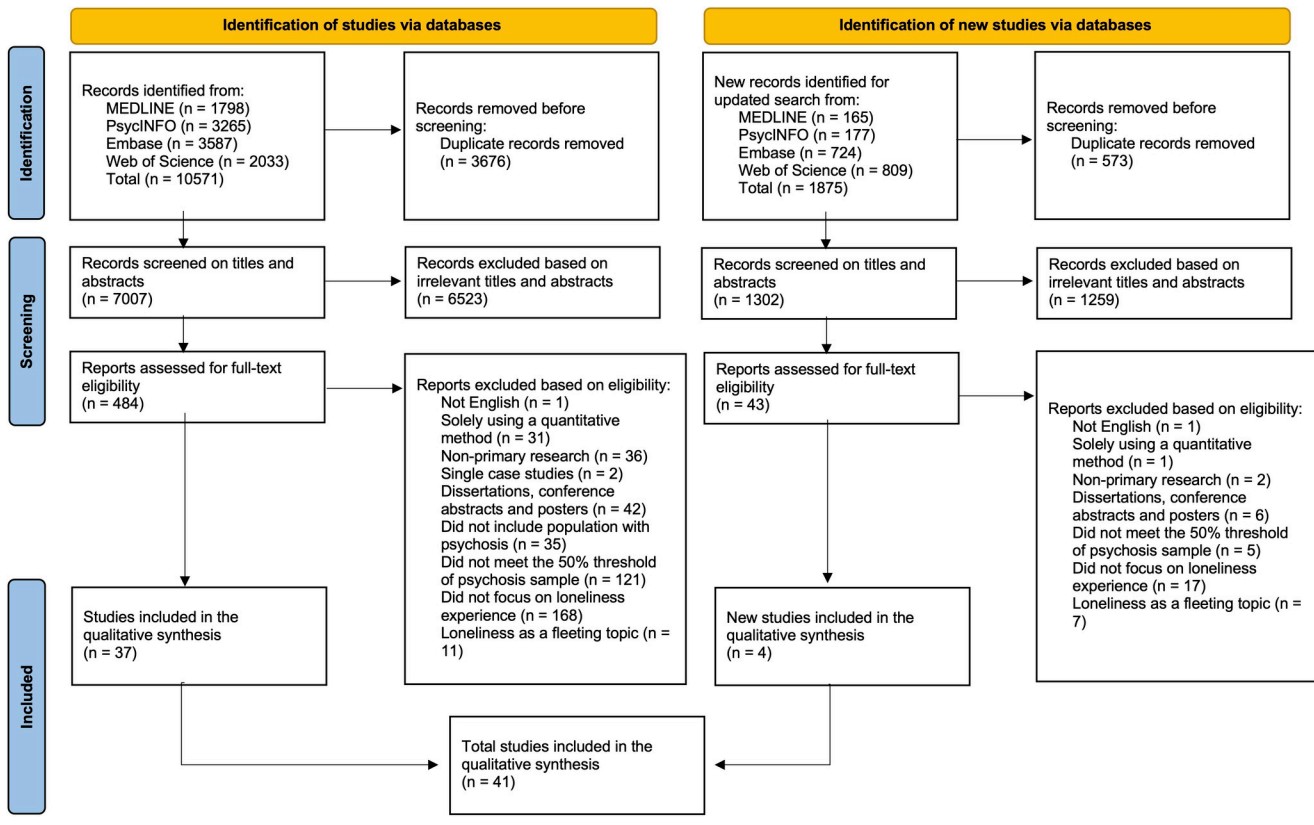

**Fig 1. PRISMA flow diagram for included studies.**

employed, with most describing use of thematic analysis (n = 16), followed by interpretative phenomenological analysis (n = 9) and grounded theory (n = 8).

## Thematic synthesis

Through a process of thematic synthesis [31] of data extracted from the above studies, we identified six meta-themes, three of which had two sub-themes, described below with illustrative quotes from included research studies. Fig 2 represents the thematic framework. S7 Appendix provides additional illustrative quotations to support this framework. Secondary interpretations from included studies are presented in quotation marks, while primary quotes from study participants are depicted in italicised text and quotation marks.

**Theme 1: Loneliness in the form of disconnection.** Most studies highlighted a strong sense of many participants feeling disconnected, characterised by feeling like outcasts who were different from everyone else and involuntarily living in "the world of the psychotic experience" [49–53]. Some participants described this form of disconnection as an "inner strangeness" and "emotional separation" that made it difficult for them to share their feelings with family and friends, and hindered progression towards their purpose in life [44, 47, 54–57]. To demonstrate, one participant described their disconnectedness as:

> "Because I know you can't tell by looking at me, so at first it won't even occur to people. But that does tend to make life very complicated, and it sometimes makes me extremely lonely. Just this week I had the feeling that I was awake and the rest were asleep. Other people haven't

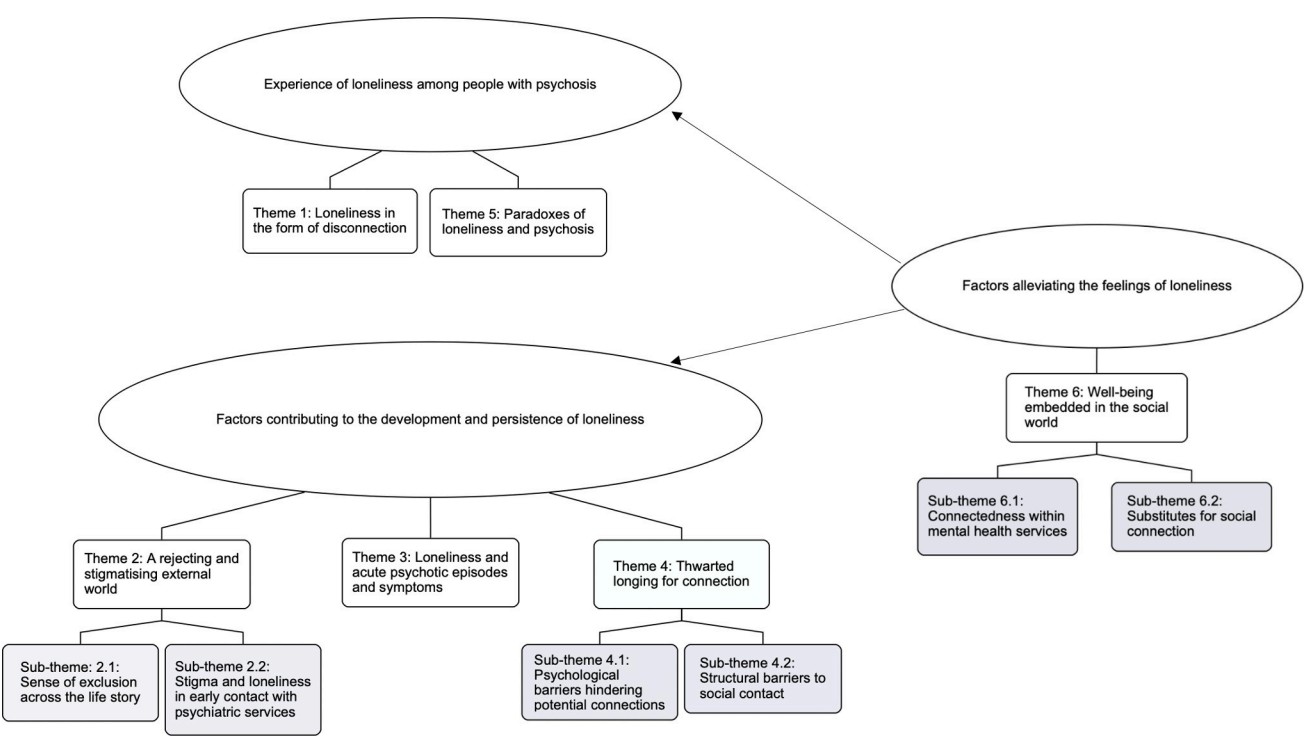

**Fig 2. Map of meta-themes and sub-themes.**

*been through any of this, they are sleeping, and they have no idea what it is like to... Do you know what I'm saying? And for me there is no way back."* [56]

Some participants highlighted the enduring nature of loneliness, featuring emotional detachment from others regardless of whether they were physically accompanied by someone [47, 48]. They described loneliness as constraining, with one participant reporting that they felt as if *"locked in a room all by yourself with no doors"* where no one could reach them [47]. Another participant described that they felt that they belonged to a "different dimension" [58]. However, other participants emphasised the feelings of loneliness could be reduced when around trusted others, for example in the context of being married [59, 60].

Some participants also commented on the feelings of emptiness and helplessness due to a sense of gradual disconnection from the others' reality, such as daily experiences in social, work and home life, and a sense of forced and painful period of stagnation and absence from "normal life" [42, 46, 48, 50, 51, 57, 61]. A participant expressed mourning for "the gap between their old life, before the onset of the disease, and their current life" [62]. Their narratives suggested a fear of *"falling behind"* [63], *"missing out"* [57], *"being a burden"* and as someone who had to *"beg for social companionship with people"* [46].

*"Everything seems to go too fast. I don't fit in because I am not really studying anymore anyway. Writing e-mails, making new friends—it is such hard work. How to put it... I haven't seen them [friends] for a year and I just have the feeling that I don't belong. I don't even know if I want to belong."* [56]

Moreover, several participants expressed feeling unsupported, misunderstood or underappreciated by confidants or loved ones, which could amplify their suffering from loneliness [48, 64, 65]:

*"Loneliness for me is when I feel like I can't tell anybody that I have an issue, I can't confide in anyone. . . Like nobody cares, that's what loneliness is to me. Like nobody understands or cares that I'm going through a rough time."* [48]

One participant expressed that despite having supportive friends and family, he still felt different as he had *"different moods than my surroundings, my friends, or my family or something"* [44].

Some studies also conveyed a sense of loneliness at an existential level, in which participants felt that they had lost their self-identity and were alienated even from their own feelings [44, 49, 51, 55, 58, 62, 66]. They experienced loneliness as an "invisible companion" that brought pain [49].

In one study, some participants expressed that they felt calm throughout COVID-19 knowing that they were not alone in their struggle with loneliness, with one participant stating that *"because so many people have been affected and experienced being in confined spaces, it felt like they might understand what I am going through, and it has helped me feel less isolated. It is like I found some comfort in knowing that others can relate to what I am going through because of COVID-19"* [45].

A participant described a sense of not belonging as his *"greatest tragedy"* because he felt destined to live without love, which he considered as a fundamental quality of humanity [62]. Some expressed the feelings of loneliness in terms of being forsaken by God or that they *"belonged with the animals"* instead [47, 54, 55, 62]. However, a study comparing different age groups found that younger people who had experienced shorter periods of psychosis were more hopeful in terms of anticipating the building of a loving relationship in the future, as compared to people who had experienced longer periods of psychosis [67].

**Theme 2: A rejecting and stigmatising external world.** This theme reflected the feelings of loneliness caused by rejection and stigmatisation from family, friends and the general society [41, 43, 47, 57, 62, 66, 68–71]. Participants in multiple studies described not being welcomed by others, or being discriminated against. A few participants reported being labelled as *"a loose cannon"*, a *"lunatic"* or a *"nutcase"* who could be perceived as dangerous to society [61, 64, 70, 72]. Other participants described feeling like a "scapegoat" who could be looked down on as someone who *"was 'more ill' than anyone else,"* [69] or being *"thrown out like garbage"* and *"ridiculed"* as they felt that others were ashamed of them [62, 72]. Some described that *"friends were frightened"*, *"embarrassed"* and *"weirded out"* by their diagnosis, especially when psychotic symptoms were very apparent, which then prevented them from accessing "self-esteem enhancing" opportunities within social groups that they once were a part of [43, 64, 72].

The sub-themes below captured how loneliness was embedded in the experiences of rejection and stigmatisation across the life span and in early contacts with psychiatric services.

**Sub-theme 2.1: Sense of exclusion across the life story.** This sub-theme highlighted the pervasiveness of loneliness across participants' lifecourse even before they were diagnosed with a psychotic illness. Some participants described a lifelong sense of isolation, featuring emotional separations or a lack of social support that began early in life, such as the loss of or estrangement from parents, and that seemed to increase in frequency with age [42, 47, 49, 60, 63, 67, 73]. Specifically, loneliness appeared to accrue over time as participants felt more hopeless about their situation when negative experiences accumulated.

A few participants made sense of their psychotic symptoms by referring to past experiences of alienation and victimization, which triggered feelings of worthlessness and loneliness that they wished to escape from [44, 47, 74]. To demonstrate, one participant understood the grandiose content of his psychotic beliefs as rooted in loneliness:

*"Loneliness is the most troublesome thing throughout my whole life. Perhaps one wants life to be different, yes. I have noticed that loneliness is the reason for my thoughts. Don't know. If that's a cause, then, yes."* [74]

Some participants expressed feeling distant from other people at a young age, and described these feelings of differentness as originating from difficult childhood and/or adolescent experiences, such as the absence of parental figures, being bullied at school, online rumours, verbal and sexual abuse, and neglect [44, 47, 60, 67, 74]. Another participant described the grandiose content in his hallucinations in relation to a "wish for vengeance" provoked by social rejection and victimization:

*"I think it comes from being bullied as a child; I was always a lonely child. They bullied me. They commented on everything I did, for example, how I walked. They said; 'Why do you walk like that?'. . . I've always wanted to get back at those who did it, perhaps I long for that."* [74]

One participant described a sense of exclusion since childhood, linked to their bi-racial ethnic background, and reported that: *"people won't include me in the group so much because I'm always different. There's not a lot of biracial people in Montreal"* [44].

**Sub-theme 2.2: Stigma and loneliness in early contact with psychiatric services.** Some participants described loneliness as stemming from experiencing a lack of support from mental health services, particularly during early contact and their first hospitalization [60, 67, 75]. They described feeling lost in the *"new and strange surroundings"*, with little assistance on who to contact or how to articulate the challenges they had been facing [60]. One participant commented on how hesitant she was to speak in the setting of psychiatric service for the first time, and reflected on the inner loneliness she experienced:

*"But when I was so locked up inside myself, I felt that it might have been useful to, in a way, get some help to talk [. . .] at the time, it was a really heavy responsibility."* [60]

Several participants described feeling *"treated less-than"* due to the psychiatric label imposed on them during early contact with psychiatric services. They had since struggled to escape this self-stigmatising mentality in order to have the confidence to interact with people [44, 66, 75].

*"I still felt way removed from everybody else's perspective. I felt alone and isolated, I kind of felt like I was lower than everybody else in a certain way. . . Also it was hard for me to. . . socialize with people, because I felt kind of like I had been tainted because they [the hospital] told me I had psychosis and that I might have schizophrenia that I might be bipolar, you know [the hospital] just kept throwing different things at me so it was hard to socialize with others because it was hard for me to feel like other people had those problems. . ."* [75]

A few participants identified their first psychiatric hospitalization as one of the main factors that had separated them from their social world, in that they had lost relationships due to the stigma attached to being admitted to psychiatric services [67].

**Theme 3: Loneliness and acute psychotic episodes and symptoms.** Some narratives illustrated that acute psychotic episodes (the experience of active positive psychotic symptoms) and periods of mental health crisis seemed to be the times when loneliness became the most intense [46, 48, 53, 55, 56, 63, 67, 73, 76]. Participants explained that symptoms of psychosis, such as delusions of persecution and acute episodes of paranoia, compromised their ability to communicate and maintain social relationships, which in turn intensified their loneliness [62]. Some participants believed that relational losses and intense periods of loneliness were prominent during the early stages of treatment, untreated stages or when first admitted to hospital [64, 73]. One participant explained that he had always been close with his family, however "upon the onset of auditory hallucinations, he began to experience social *'withdrawal'* and *'estrangement'* from others", stating that:

> *"I used to have friends but once I started hearing and seeing things—I didn't really. . . communicate. . . I don't know [why]. . . I was just worried. I was very inside myself."* [73]

A few participants also described how the experience of threatening voices had reduced their ability to trust themselves as well as others, leading to social withdrawal, feelings of loneliness and a sense of disconnection [48, 53, 55–57, 64, 67, 73, 76]. Participants also feared that they would respond to voices or act "strange" in public, which led to "concerns regarding how they would appear", embarrassment, shame, and withdrawal from friends and others [48, 53, 57, 61, 77]. One participant stated that when friends started to become integrated into his experience of psychotic symptoms and nightmares, he had *"locked"* himself away [57]. A few participants described their friendships ending or being disowned by friends as a result of "unusual" behaviours and paranoia during their acute psychotic episodes [57, 58, 64,76]. One participant described how psychotic symptoms had damaged their friendships, stating that:

> *"I went to see him [friend] and then over there I was acting really weird as well and I got arrested by the police over there. It was like a really small village and it, sort of, brought shame on them. . . they [friends] found out here in London, then they just stopped talking to me."* [57]

A few participants emphasised how negative symptoms of psychosis such as anhedonia had contributed to withdrawal, hopelessness and loss of pleasure, which further intensified their feelings of loneliness and, at times, led to self-harm and suicidality [43, 47, 52, 61, 69]:

> *"I have nearly no interaction with anyone; I am just alone at home. Now everyday life annoys me. I lost my interests in everything. I am doing nothing even though bored. As a result, I am much worse, and once tried to commit suicide."* [63]

Participants described several reasons for why they felt lonely and the need to withdraw during times of crisis. A couple of participants expressed that they had limited energy and reduced pleasure in social activities; they commented that they had to use their energy to concentrate on *"being a human being"* as well as "coping with the most basic tasks of life during periods of mental hardship" [46, 47, 48]. In one study, participants described that they felt estranged by an "unpredictable" and "fragmented" understanding of their interpersonal experiences, whereby they felt disoriented interacting with people [51, 67]. A few experienced mistreatment and felt *"left behind"* during this vulnerable time [55, 67, 73], while others expressed loss of social and cognitive skills, and social anxiety as barriers to connecting [48, 56, 75].
However, several participants described that they pushed themselves to socialise and form valued social connections, often with people with lived experience of hearing voices, which

helped them manage the symptoms and overcome the sense of shame [53]. Furthermore, a few participants shared the importance and value of cultivating "strong and reliable friendships" with people who remained supportive during times of vulnerability [57, 65].

**Theme 4: Thwarted longing for connection.** Most narratives commented on a thwarted longing for acceptance and connectedness. Participants expressed a strong desire for a sense of belongingness in family and society, as well as in friendships and romantic relationships [42, 47, 48, 54, 57, 63, 69, 78, 79]. One participant described her experience of loneliness as "dreams of a relationship that had not come true" [73], while another described his "craving" for an escape from loneliness as akin to a sense of hunger:

*"I've been hungry. . . for life, for hope, recovery, to find my place in the world, hungry for love. I mean hunger is a basic need that needs to be. . . it's to be. . . everybody needs. . . hungry for attention, everything."* [55]

Nonetheless, several studies highlighted how challenging it could be for participants to connect with people, with barriers including inexperience or insecurity around interacting with others, or a sense of "ineptness" in relation to intuitive social knowledge [42, 47, 51, 66]. Some narratives described the sentiment of lacking "a repertoire of responses to deal with feelings of loneliness", as participants were unsure of the specific actions to connect with other people [42, 47, 54]. In the sub-themes below, we captured the psychological and structural barriers faced by participants in alleviating loneliness.

**Sub-theme 4.1: Psychological barriers hindering potential connections.** This subtheme contained descriptions of psychological difficulties that were felt to exacerbate loneliness and prevent people from attaining meaningful social relationships. Several participants explained their loneliness as originating from social anxiety and nervousness, as they were worried other people could notice changes in their personality or patterns of socialising following the onset of psychosis, which led to insecurity and low confidence in their ability to interact with others [48, 57, 67]. Some participants described finding it difficult to initiate social interaction "spontaneously", and their social contact was characterised by distancing themself from others and careful planning of interactions [47, 51]. This created major barriers in building functional connections with others. To illustrate, one participant reflected on the dilemma she faced in maintaining contact with her friends:

*". . . if I ever undervalued friendship, I don't now, you know? Definitely don't undervalue it. It's really tough because it puts you under a lot of pressure as well to maintain contact with people and you can't always deal with that pressure all the time doing that all the time. . ."* [76]

A few participants found it challenging to sustain romantic relationships because the experience of psychotic symptoms could contribute to distrust, secrecy and difficulty communicating emotional intimacy [67]. One participant reflected on her unreadiness to bond with a romantic partner despite longing for the connection:

*"The thing is, right now my emotions are sort of extinguished, they're the opposite of vibrant. [. . .] It's just that some things don't reach me, or they bounce off. [. . .] Because if you're falling in love, those emotions have to be there. They just have to. . ."* [67]

A few studies highlighted what appeared to be cognitive barriers participants faced in comprehending social behaviours, meanings and expectations [47, 51]. This could result in

inappropriate responses that led to further rejection and disconnection from others. To demonstrate, one participant reflected on his attempt to initiate social contact whilst finding great difficulty making sense of the social world, leaving him with a sense of loneliness and social failure:

*"I changed my mind when I got there, so to speak. [. . .] I didn't dare make a mistake, because it would be, like, wrong to meet someone by just going up and ringing the doorbell. So I get there and decide. . . 'nooo. . . I don't f\*\*\*\*\*\* dare, I don't dare do a thing, because I am so damn weird right now.'"* [51]

**Sub-theme 4.2: Structural barriers to social contact.**   Some studies highlighted structural obstacles that exacerbated loneliness among people with psychosis. Due to psychotic symptoms or other functional impairments associated with their diagnosis, many were economically inactive and dependent on disability allowance or support from family. Participants longed for social engagement yet felt trapped by financial insecurity, which made it challenging for them to meet new people or sustain relationships in public social spaces [46, 48, 67].

*"I don't want to bring up [hanging out with my best friend] because I don't have money to go out and I feel like I would only be able to say, 'Hey, do you want to just come hang out and sit around my apartment.'"* [48]

Several participants also emphasized the negative connotations of disrupted employment, poverty and mental disorders, which were labelled as undesirable in social relationships. This added to the burden of diagnosis and became "a determinant of multiple, cross-sectional stigma" that aggravated rejection, disconnection and loneliness [46, 64, 67].

*"It's not only about the money. It's about status, social status. [. . .] But if you appear as a bloody loser and think 'I have no job, I have no money, I have no health, I have no sense in my head . . .' then it's, of course, meant to fail. You might as well go to bed. . ."* [46]

**Theme 5: Paradoxes of loneliness and psychosis.**   Many narratives depicted a contradictory desire for solitude alongside a profound longing for connection [42, 54, 55, 73]. To demonstrate, one participant expressed his disinterest in connecting with people, then made a paradoxical statement about his sense of loneliness:

*"I really can't say that I want to be a part of anybody else's life, or something like that, you know what I mean? [. . .] At times, I feel like, uh, just hearing voices, stuff like that, maybe I feel like I'm. . . uh, I guess maybe I get lonely at times."* [54]

Some participants described needing to withdraw from others due to past experiences of victimization, which left them hurt as if they were "stranded in the dark" and reinforced their perception of relationships being "imminent and existential threats" [42, 55, 60, 69, 71, 73]. They felt lonely and craved interpersonal closeness yet felt anxious about the potential risk of bonding with people. One participant reflected on her dilemma of faith in others:

*"It didn't go so well with friends in the past. Eh, been bullied a lot. By people who used to be my friends. . . So, it is, in a way, a bit hard to be able to, like, trust people. Although. . . I really want to."* [60]

Participants described how stigmatisation and rejection inflicted on them *"a kind of paralyzing sadness"* that aggravated their feelings of loneliness, and further discouraged them from social interactions despite a yearning for companionship [41, 48, 56, 59, 63, 67, 68, 71, 80].

> *"I think until society changes some of its perceptions about what it's like to be a schizophrenic, what the reality is for most of us, you know they're not sort of raving homicidal lunatics all the time, that it's going to be difficult because it's against that backdrop that you drop the bombshell, so to speak, about what your diagnosis is. . ."* [59]

Narratives also emphasized the role of participants' internalized stigma and feelings of shame surrounding their diagnosis, which deterred them from engaging socially [42, 48, 49, 59, 68, 71, 76, 80]. They did not want to burden or upset others with their psychotic symptoms [49, 57], neither did they want to build up their hopes about social relationships just to be rejected and let down again [67, 72, 80]. It was therefore less anxiety-inducing for some participants if they devised a strategy that permitted themselves to have no expectations and therefore no disappointments; a mentality in which they "hardened themselves to the experience of not belonging" through establishing a self-protective distance [54].

> *"This disease [schizophrenia] is rather special, no one is willing to socialize with people with this disease. So, I'd rather stay alone, lest others disgust me."* [72]

Multiple narratives highlighted a vicious cycle of social withdrawal, in which participants distanced themselves from others because of psychosis, yet found that being alone could be detrimental to their mental health whilst also perpetuating a sense of difficulty in sustaining social interactions [41, 46, 50, 54, 56, 76]. Some were accustomed to being socially withdrawn and had become progressively more uneasy around other people [56, 63, 67]. To illustrate, one participant reflected on their experience at school after being diagnosed with schizophrenia:

> *"I'd always been withdrawn from schoolmates. I was extremely ambivalent about being with somebody. When I was alone, I felt comfortable; at the same time, I was so lonely!"* [63]

To protect themselves from rejection and stress associated with trying to gain a sense of belongingness, participants created an emotional distance from others through "building a wall" around themselves, which included strategies such as social withdrawal, partial or non-disclosure of disorder, and showing "an attitude of indifference" [43, 54, 55, 59, 73]. A few participants "tested the water" with people in their social circle to assess the risk of rejection, through initiating a general conversation about mental health difficulties [59]. These provided them with some sense of security in the threatening social world.

**Theme 6: Well-being embedded in the social world.** This theme influenced themes 1–5 as a mitigating factor. It highlighted how forming social bonds with others could help people with psychosis cope with their feelings of loneliness and symptoms, as social interactions helped preserve "a sense of normality" in their lives and offered distraction from illness-related rumination [57, 76]. Participants endorsed "continuity" and "reciprocity" as two core qualities they valued in relationships, be it with friends, family, romantic partners or mental health professionals [43, 59, 73]. These features of social connection helped generate positive feelings, cultivate belongingness and facilitate recovery [54, 59].

*"[Belonging means] constantly making contact with friends, no matter how few you have, family, girlfriend, whatever. You have to maintain those things in order to have a belonging, at least I feel so. Those are the things that make up a human."* [54]

Some participants highlighted the positive role of friendships in providing support to reintegrate into society or resume everyday activities that could be challenging to initiate alone, such as playing sports or going to the movies [57, 76]. Participants also emphasized the importance of "being a part of something" in reducing the feelings of loneliness, such as getting a job and participating in recreational activities, through which they were able to form social connections, achieve their goals and find purpose in life [48, 54, 79].

*"It's like a sponge. Sometimes I'm filled with water and sometimes I'm not, just depending on how tightly the voices are squeezing at me. When I'm alone, it's usually at it's worse. I try to, as much as possible, try to integrate myself back into society because that's the only thing that can fix it."* [54]

Some participants described the importance of having agency over how to live the life of someone with psychosis in solitude [48, 58]. They expressed that unwanted solitude could exacerbate loneliness, but having control over the context in which they could be alone or connect with others when feeling unwell enhanced their sense of well-being [43, 46, 48, 58]. To illustrate, one participant reflected on her decision to be alone temporarily:

*"I've been quite alone these past six months, I haven't felt like seeing anyone. It's not that I have become more lonely than I was before, it's more that I've been feeling down and haven't wanted to see anyone."* [46]

In the sub-themes below, we discussed participants' attitudes towards establishing social connections within psychiatric services, and their substitutes for social contact.

**Sub-theme 6.1: Connectedness within mental health services.** While participants valued the opportunity to bond with family, friends and community members who did not identify as having a mental health problem, many emphasised the importance of fostering trusting relationships with people who had similar experiences [43, 54, 57, 68, 72, 77]. Participants commented on how crucial talking therapy and peer support groups were for diminishing loneliness and promoting recovery [52, 73, 75]. These helped them feel empowered through having a safe space to share their personal experience and be supported "as individuals and not just one in the crowd" [68]. They appreciated how connecting with people who had similar experiences helped them "move past feelings of abnormality and self-stigma", build up interpersonal skills, offer hope to get through hard times, and cope with the psychological distress of psychosis [48, 54, 75, 80, 81]. Participants described that being around others whom they could relate to helped promote a sense of purpose and meaning in their life [54].

*"Stuff goes on in my life. And I can discuss the problems in [support] group. . . Sometimes I just feel depressed in my head. . . or just not good. It helps me to sit in there. It helps just to sit and listen. . . and I honestly like to help people out. . . The clients have their problems, and [I] give them advice. . . Give them feedback. . . I get a lot of feedback and it helps me."* [73]

Some participants described that "they would feel lonely, 'lost' and 'adrift' without their treatment providers", as they developed meaningful bonds that promoted a sense of well-being in the absence of friends or family members [73, 80]. Participants emphasized how valuable it

was to form secure therapeutic relationships with healthcare professionals, as they felt heard and understood [68].

> "[. . .] I had a lot of crises in my life and he [psychiatrist] was there for me. . . He provided me with an intimate relationship. . . [and] a support structure." [73]

In addition, participants spoke of the importance of "a sense of agency and choice" over the types of social groups they associated with, as they sought to interact with people with or without psychosis at different times in life depending on the types of support they needed [43, 57, 77]. To demonstrate, one participant described the differences in her attitudes towards group memberships when symptoms worsened compared to when they improved:

> "I think it was good at the time [when psychosis worse] to be amongst others that experience the same thing and you know realise I'm not alone erm however some of the stories that I hear in there [peer support group] I just, you know, I don't know, I haven't felt any benefit from keeping going. I haven't been going as often as I did [. . .] I know that people [from that group] are in that state permanently and that's lots of sad stories." [43]

A few participants explicitly expressed a desire to disengage from the psychiatric community as they no longer identified with people who had mental health conditions [46, 47]. Some of these participants acknowledged the benefits of peer support and community, but valued their social identity in the wider community outside mental health services:

> "You know it feels really good to be around 'normal' people. . . someone that's a non-alcoholic or whatever. When you're out there in the real world, working, you're not in this protective meeting where everyone's like you. You're not in group therapy and everyone has the same mental health diagnosis. You're out mixing in the community and being human. . . You're just you, no label." [48]

**Sub-theme 6.2: Substitutes for social connection.** Some participants described a "positive form of loneliness", as they substituted social interaction with solitary interests such as painting and listening to music [46, 73]. Other participants who were able to engage in activities away from home, such as grocery shopping or running errands, expressed finding a sense of purpose that protected them against the feelings of disconnectedness [48]. Several participants described physical proximity to others in public spaces as useful in diminishing loneliness [48], and one participant regarded smoking as a *"friend"* and an "adequate replacement for social contact" [73].

Three participants from a study reported not feeling lonely, as they relied on hallucinations for companionship [47]. Similarly, two other studies highlighted the positive nature of supportive content in some participants' hallucinatory voices [74, 76]. They appeared to derive social well-being from their hallucinations as they referred to the voices as fulfilling the "friendship role" [76] and being "substitutes for loneliness and longing" [74]. Among these narratives, participants with more depleted social connections appeared to place a greater value on their relationships with their voices than those who described more social support [76]. To illustrate, one participant reflected on her decision not to take medications so that she could maintain "contact" with the hallucinatory content of her ex-partner:

> "There was a part of me who really wanted to believe in this, that it was true, so I didn't renew my prescription because I was feeling so bad, because at the same time this was my only link to him." [74]

## Discussion

### Main findings

This meta-synthesis of 41 qualitative studies identified six meta-themes and six sub-themes, conveying the nature of loneliness among people with psychosis, the pathways between psychosis and loneliness, and how these could be mutually reinforcing. This thematic framework highlighted a sense of belongingness as the central facilitator of recovery from psychosis.

Our analysis identified two themes that mapped to our first research question about how individuals with psychosis experienced loneliness. We found that people with psychosis experienced loneliness in the form of disconnection from self and the others (theme 1), and they described a paradoxical yearning for closeness and distance, driven by fear, anxiety and self-stigma (theme 5). Our second research question related to identifying the factors people with psychosis perceive as contributing to the development and persistence of loneliness, and we identified three themes that addressed this. Individuals with psychosis consistently highlighted their experiences of rejection and stigmatisation as the source of loneliness, particularly where there was early contact with psychiatric services, which could aggravate loneliness (theme 2). Some also described acute psychotic episodes and periods of mental health crisis as constituting the most intense periods of loneliness in their lives (theme 3). Participants also shared the various psychological and structural barriers that could hinder social contact, including social anxiety, insecurity and the label of low socioeconomic status associated with psychosis (theme 4). Our analysis identified one theme (theme 6) outside the scope of our research questions: how social connections, when trusting and reciprocal, could build belongingness and reduce the feelings of loneliness. This theme influenced all other themes in our analysis by operating as a mitigating factor (i.e., influencing the onset and/or perpetuation of loneliness in people with psychosis), which was highly relevant to the exploration of the experience and perpetuating factors of loneliness among people with psychosis.

We did not find any qualitative accounts that described how the experience of loneliness might vary with respect to the different types and phases of psychotic disorders, the content and nature of hallucinations or delusions, the balance of positive versus negative symptoms, and the specific patient characteristics (e.g. ethnicity, socioeconomic status, sexuality, and any other characteristics of interest). These are evidence gaps for future qualitative exploration.

We did not identify any patterning of themes based on age, gender or cultural background, but it was challenging to examine this possibility without access to the primary data. Some participants reported that their sense of emotional detachment increased with age, but experiences of loneliness appeared to be prevalent across all stages of psychosis.

### Findings in the context of other studies

To our knowledge this is the first qualitative meta-synthesis of studies exploring the experience of loneliness among individuals with psychosis. Our findings suggest that for people with psychosis, loneliness is prominent across social, intimate and public spaces, and has features and origins that are at least in part specifically linked to psychosis [82, 83]. People can feel disconnected due to challenges in forming reciprocal confiding relationships and sustaining emotional attachment, as well as stigma and social rejection, which altogether impact their ability to build meaningful relationships and induce loneliness [2, 4]. Our theme capturing the experience of disconnection highlights the existential and ontological contexts of loneliness in people with psychosis, which they describe as a sense of emptiness, shamefulness, and abandonment [84–86].

Our study findings are congruent with those from a systematic review of quantitative studies, which concluded that loneliness among people with psychosis is mediated by anxiety, internalized stigma, perceived discrimination and low self-esteem [10]. The accounts of thwarted desire for connection due to social anhedonia, affective flattening and difficulty understanding social behaviours also support the link that has been reported between loneliness and negative psychotic symptoms [87]. In addition, there were many narratives of experiencing loss, abandonment or neglect, including early in life, suggesting that loneliness, interpersonal difficulties and psychosis may have common roots [17, 25]. Our results support the existing research suggesting that paranoia can interact with past and current experiences of rejection and discrimination to reinforce self-protective social avoidance, hypervigilance to social threats, self-stigmatisation and the associated negative emotions, which contributes to a vicious cycle of loneliness and psychosis as described by the "reaffiliation motive" theory [88–92].

Similar to our findings, a narrative synthesis of qualitative studies exploring what recovery means for people with mental health problems conceptualised personal recovery as involving connectedness, hope about the future, and the redefining of a positive sense of self [93]; peer support was highlighted as a valuable therapeutic asset in reducing paranoia by building up one's personal worth and life purpose [94]. These data highlight the integral role of social connection in integrating individuals with the community and bringing about a sense of normality and purpose in life, as well as the importance of having agency over group memberships. These findings align with the social identity theory, which postulates that social identification may reduce the symptoms of paranoia by increasing perceived control, trust and self-esteem [43, 94, 95]. Our findings regarding the psychological and structural barriers to social engagement suggest that some direct and indirect approaches in loneliness interventions, as identified in two scoping and systematic reviews on the interventions for subjective social isolation in people with mental health problems [21, 22], may be beneficial for individuals with psychosis depending on their needs. For example, the chronic nature of loneliness and its close links to symptoms and cognitions among people with psychosis suggest a need to consider direct approaches focused on social skills training and challenging unhelpful cognitions (e.g., lack of interpersonal trust), which is in keeping with previous studies on the current challenges and gaps in loneliness interventions among people with psychosis [10, 12]. These intervention approaches can benefit from offering people with psychosis the support in connecting with peers who have similar experiences, and improving the quality of their pre-existing relationships with friends and family. In addition, our findings on the links between stigma and loneliness highlight the importance of tackling social challenges presented by low socioeconomic status associated with psychosis (e.g., improving education and employment opportunities), as well as the significance of community-wide initiatives in addressing stigma of psychosis and facilitating integration into the wider community. It is important to note that educational campaigns focused on improving public mental health literacy might not reduce stigma if they promote the idea of "otherness" in relation to people with psychosis, therefore anti-stigma interventions could benefit from a focus on optimism about recovery and challenging the perceived dangerousness of psychosis [96]. Theme 3 corroborates quantitative and qualitative findings suggesting that addressing psychological and structural barriers associated with loneliness and social needs in the early stages of illness can be beneficial to recovery, particularly as loneliness may trigger the symptoms of psychosis [20, 97].

## Strengths and limitations

This meta-synthesis used a well-established reviewing approach [31, 34] to synthesise existing primary qualitative evidence in order to address important questions concerning the nature

and pathways of the experience of loneliness among individuals with psychosis. Such evidence has potential to inform the development of interventions to alleviate loneliness among people with psychosis. Our search terms on loneliness were broad, involving conceptually overlapping or closely-related concepts, thus establishing a rich dataset. Threats to validity and reflexivity were considered and addressed through regular discussions with the multidisciplinary review team.

However, our review was limited in several ways. Firstly, results of this meta-synthesis are restricted by the methodological quality of the included research, including limited reported reflexivity in the original papers. In addition, aggregating the results of studies conducted in a wide range of settings with different theoretical frameworks and analytic approaches inevitably leads to some loss of context and nuance in data. We made efforts to be mindful of this by situating authors' interpretations in the context of the corresponding primary quotes from research participants. The involvement of lived experience researchers in reviewing the paper and providing commentary also helped address reflexivity and provide more confidence to the interpretations.

Another limitation of this review is that, due to resource and time constraints, a sensitivity analysis and an assessment of the certainty of evidence (e.g., the GRADE-CERQual approach) were not conducted for each individual finding. Therefore, the findings of this review should be interpreted with caution.

We acknowledge that data were primarily generated from high-income countries, especially the United Kingdom and the United States. Some papers did not specify participants' demographic characteristics or the types of psychotic disorders they had, which reduced our capacity to investigate patterns of qualitative themes by age, gender, ethnicity, and context of psychotic symptoms. Finally, only four included articles explicitly aimed to investigate loneliness among people with psychosis. We therefore relied on materials extracted from a broader body of work exploring people's general social experiences.

### Clinical implications and future research

This meta-synthesis highlighted the centrality of loneliness for many people with psychosis, as well as the mutually-reinforcing interaction between loneliness and the positive and negative dimensions of psychotic symptoms. This suggests a need for clinicians and researchers to collaborate with people with lived experience of psychosis to tailor loneliness interventions to their particular needs and experiences, while actively listening to their feelings of loneliness, and incorporating routine measures of loneliness in mental health services to promote understanding of service users' relationships with the social world. Other key suggestions to mental health professionals include (i) exploring people's experiences of positive and negative psychotic symptoms, and their impacts on social relationships, (ii) encouraging reciprocity and continuity in therapeutic relationships. Findings in this study also emphasised the importance for clinicians to address factors that might be perceived as threatening or overwhelming in the settings of group interventions or psychiatric services, including the externalized and internalized stigma associated with the labelling of psychosis.

In addition, the centrality of a sense of belongingness in alleviating loneliness suggests a need to develop socially-focused interventions that connect individuals with communities that they hope to develop social group membership and sense of self in. Signposting to peer support from people with similar lived experience and social prescribing are potential approaches in linking individuals to social and emotional support, thus bringing about a sense of purpose and normality through social connections. Loneliness intervention development would benefit from community-wide campaigns that focus on addressing the marginalisation of people with

psychosis, and reducing the stigma associated with psychosis. Our findings also highlight the potential importance of challenging maladaptive social cognitions in alleviating loneliness, through social skills training and challenging internalized stigma associated with psychosis. In conjunction with the evidence of high self-esteem mediating the link between social identification and reduced psychotic symptoms, such as paranoia, avolition and social withdrawal [98, 99], cognitive behavioural treatment for self-esteem could be beneficial for people with psychosis.

Future studies are required to develop co-produced interventions for psychosis whose feasibility, acceptability and effectiveness can then be tested. Only four included articles explicitly aimed to investigate the experience of loneliness among individuals with psychosis, therefore further qualitative research with a specific focus on the nature of loneliness and sense of belonging, and the influences on these, is needed both among individuals with a psychotic diagnosis and those with a high risk of psychosis. It is also important for future research to probe the roles of social cognitions and stigma in the context of loneliness and psychosis. Such data can deepen our understanding of the various pathways linking loneliness and psychotic symptoms. Given the evidence gaps we identified, there is a need for empirical studies to explore how the experience of loneliness varies with respect to factors such as stages of illness, types of psychotic disorders, or specific patient characteristics. Future qualitative interview studies should use purposive sampling to ensure that people from a range of backgrounds, stages of recovery and marginalised groups are included. This would allow us to gain an in-depth understanding of loneliness and how it is influenced by societal and social barriers. Once these phenomenological knowledge gaps are addressed, it will be appropriate to develop a conceptual model that captures the complexity of the relationships between loneliness and psychosis.

## Conclusions

This meta-synthesis of 41 qualitative studies established six meta-themes and six sub-themes illustrating the experience of loneliness among individuals with psychosis. It was highlighted that participants experienced enduring disconnection and loneliness, embedded in their experiences of rejection, stigmatisation and victimization. Loneliness was particularly prominent during periods of active psychotic symptoms and acute psychotic episodes, which at times required crisis intervention. People with psychosis described a paradoxical longing for distance from others yet also longed for reciprocal connections. They felt stuck in a self-perpetuating cycle in which social withdrawal, loneliness and psychosis were mutually reinforcing. The centrality of loneliness we observed in the daily experience of individuals with psychosis suggests a need for tailored loneliness interventions to meet their needs for belongingness and social connection.

## Supporting information

**S1 Appendix. PRISMA checklist.**
(DOCX)

**S2 Appendix. ENTREQ checklist.**
(DOCX)

**S3 Appendix. Search strategy.**
(DOCX)

**S4 Appendix. List of studies excluded at full text screening.**
(DOCX)

**S5 Appendix. Table presenting characteristics of eligible articles.**
(DOCX)

**S6 Appendix. Evaluation of study quality according to the Critical Appraisal Skill Programme (CASP) qualitative checklist.**
(DOCX)

**S7 Appendix. Themes, sub-themes and illustrative quotations by authors and participants contributing to meta-synthesized themes.**
(DOCX)

## Acknowledgments

Thank you to the UCL Loneliness and Social Isolation in Mental Health Network Co-Production Group (CoG) for their support with this research.

## Author Contributions

**Conceptualization:** Cheuk Ying Lee, Nafiso Ahmed, Brynmor Lloyd-Evans, Sonia Johnson.

**Data curation:** Cheuk Ying Lee, Nafiso Ahmed, Brynmor Lloyd-Evans, Sonia Johnson.

**Formal analysis:** Cheuk Ying Lee, Nafiso Ahmed, Sarah Ikhtabi, Leonie Richardson, Alexandra Pitman, Brynmor Lloyd-Evans, Sonia Johnson.

**Investigation:** Cheuk Ying Lee, Nafiso Ahmed, Sarah Ikhtabi, Phoebe Duffett, Yazan Alhilow, Leonie Richardson.

**Methodology:** Cheuk Ying Lee, Nafiso Ahmed, Brynmor Lloyd-Evans, Sonia Johnson.

**Project administration:** Cheuk Ying Lee, Nafiso Ahmed, Brynmor Lloyd-Evans, Sonia Johnson.

**Supervision:** Nafiso Ahmed, Brynmor Lloyd-Evans, Sonia Johnson.

**Validation:** Cheuk Ying Lee, Nafiso Ahmed, Sarah Ikhtabi, Leonie Richardson.

**Visualization:** Cheuk Ying Lee.

**Writing – original draft:** Cheuk Ying Lee.

**Writing – review & editing:** Cheuk Ying Lee, Nafiso Ahmed, Sarah Ikhtabi, Leonie Richardson, Alexandra Pitman, Brynmor Lloyd-Evans, Sonia Johnson.

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
