## [Decision Letter · Decision Letter 0]

4 Oct 2024

PONE-D-24-33999The experience of loneliness among people with psychosis: qualitative meta-synthesisPLOS ONE

Dear Dr. Ahmed,

Thank you for submitting your manuscript to PLOS ONE. After careful consideration, we feel that it has merit but does not fully meet PLOS ONE’s publication criteria as it currently stands. Therefore, we invite you to submit a revised version of the manuscript that addresses the points raised during the review process.

The topic of your manuscript is highly interesting and holds great potential. However, in order for us to consider it for potential publication, revisions addressing the reviewers' comments will be necessary.==============================

We look forward to receiving your revised manuscript.

Kind regards,

Vittorio Lenzo

Academic Editor

PLOS ONE

Journal Requirements:

“SJ, AP and BLE receive salary support from the Loneliness & Social Isolation in Mental Health Research Network, which was funded by UK Research and Innovation (Grant reference: ES/S004440/1) and is now funded by the NIHR UCLH BRC. Their support is gratefully acknowledged.”

5. As required by our policy on Data Availability, please ensure your manuscript or supplementary information includes the following:

Reviewers' comments:

Reviewer's Responses to Questions

**Comments to the Author**

1. Is the manuscript technically sound, and do the data support the conclusions?

Reviewer #1: Yes

Reviewer #2: Partly

2. Has the statistical analysis been performed appropriately and rigorously? 

Reviewer #1: Yes

Reviewer #2: N/A

3. Have the authors made all data underlying the findings in their manuscript fully available?

Reviewer #1: Yes

Reviewer #2: Yes

4. Is the manuscript presented in an intelligible fashion and written in standard English?

Reviewer #1: Yes

Reviewer #2: Yes

5. Review Comments to the Author

Reviewer #1: Comment

Overall Thank you for this well executed, highly relevant paper. I support its publication. Below are some suggestions.

The paper is organised around two research questions: the experience of loneliness, contributory factors to loneliness, in psychosis. However, the Data Analysis and Discussion sections do not directly address these questions. The paper would be stronger if these were directly addressed.

Having meta-synthesised the themes how to these address the RQ’s. This might necessitate a more inductive/deductive analysis of the meta-synthesised themes.

To my mind there are two missed opportunities- one to propose a model of loneliness in psychosis- which is a fundamentally different experience that that of loneliness in the general population, which captures all the paradoxes inherent in psychosis and loneliness- which allows for the complexities in designing anti-loneliness interventions for this group. This is well captured in reference 44- “this study suggests nuance to the application of the Social Cure for a highly stigmatised, yet fluctuating, condition such as psychosis.”

two to create a visualisation which will capture this experience powerfully.

It would be interesting to address the RQ’s visually. How do the codes and subthemes cohere around the RQ’s. Figure 2 and S4 for instance, while accurate do not really address this.

Abstract This seems clear and well described. The methods subsection can include whether PRISMA guidelines were followed, what the quality appraisal tool was. This guideline may be helpful here

Competing interests The authors are funded by a loneliness charity. This is important.

Data Availability Where are the data available?. For instance, if a researcher wished to re-analyse the data is the extracted data available?

Introduction The introduction is clear and well written. It gives a good overview of the loneliness literature. However, it could be more specific. Considerable research has been done in psychosis/SMIs and loneliness, the specificity of this experience, and the challenges in addressing it. The introduction would be considerably stronger if it focused specifically on psychosis/loneliness. The introduction could really start on Pg4 Line 74 “Two Systematic reviews…”

The authors describe the challenge well “The complexity of relationships between loneliness and psychosis suggests that approaches to addressing this need to draw on an understanding of the distinctive features of experiences of

loneliness among people with psychosis”. However, in my opinion they could spend more time identifying these distinctive features in the Introduction

Reference 25- In one line, and one reference, the authors discuss loneliness interventions and psychosis. This is a substantial area of literature with considerable evidence already gathered, which could be further explored, especially the challenges inherent in the area- how do support someone to socialise who experiences paranoia, social anxiety, anhedonia or social withdrawal.

Many, many researchers are interested in reducing loneliness in psychosis. The question, which this review is aiming to address, is how?

Methodology This section is well described

Thematic Synthesis Although this is a perfectly reasonable approach to use, I wonder whether it would be supported by a more deductive approach, namely structuring the codes around the two RQs: the experience of loneliness, contributory factors loneliness in psychosis. This would necessitate two analyses rather than one. I’m loathe to suggest the authors re-analyse the data but then a better way of addressing the RQs in the Results and Discussion. Which themes/subthemes best address each question? Can this be done graphically?

Results These are clearly described

Discussion This is clearly described but, as noted above, the RQs aren’t addressed directly. For instance, based on these data what should a direct anti-loneliness intervention comprise? The authors cite Ref 24, but we’ve already established that the evidence base for this is poor.

There is room here to look at Social Identity Theory, or other social theories of belonging.

Indirect, public health interventions are important but often have paradoxical outcomes of increasing stigma. For instance. https://doi.org/10.3109/09638237.2015.1057327

Similarly, the Clinical Implications section notes some relatively generic advice, not specifically based on the data on this paper, and not addressing the challenges highlighted by Ref 24, 25, 44

Future Research This is a little generic, and does not seem particularly informed by the data in the study. Specifically, what does the data suggest is the next step in psychosis/loneliness research.

There is an established evidence base for self-esteem interventions.

“Further qualitative research with a specific focus on the nature of loneliness and sense of belonging, and the influences on these, is needed both among individuals with a psychotic diagnosis and those with a high risk of psychosis.” That’s this study.

There is an added level of depth which could be drawn from the data to address the “nature of loneliness” and “the influence on these”

Reviewer #2: Thank you for inviting me to review this very insightful review. This is topic requiring further research and development of co-designed interventions, so I think this review provides a good platform for that work. Much of my feedback is methodological, in the spirit of demonstrating the quality of the review process. I am conscious that you refer to meta-synthesis but I would suggest qualitative evidence synthesis (QES) as the preferred term, within which thematic synthesis is one of the more common approaches. So, if I refer to QES you can apply that to “meta-synthesis”

First and foremost, I would urge the authors to consider ENTREQ reporting guideline (https://www.ncbi.nlm.nih.gov/pmc/articles/PMC3552766/) available through the EQUATOR network. The EPOC template for reporting Qualitative Evidence Synthesis (QES): https://zenodo.org/records/10050961 Is also very useful for preparing your review report.

Using more appropriate reporting guidance would help to address some of the omissions I identified in the paper, for instance, why thematic synthesis? (as opposed to framework synthesis or meta-ethnography).

You have used literature to support your methodology (refs 31-35) but some of these are dated and there has been huge methodological advancements since. I would strongly urge you to review more contemporary literature in this area to inform your review stages.

In terms of quality appraisal, CASP is admittedly quite difficult to use but there would be a strong feeling in the community that the studies should not be scored as it remains quite subjective. Usually, a “yes” “no” or “unclear/can’t tell” format is used.

There is no reference to the confidence in your findings outside of the limitations of the primary studies. GRADE CERQual is commonly used now- I think it should be acknowledged as a limitation that there was no assessment of confidence, nor sensitivity analysis conducted for this review.

Theme development (please avoid term “emerge”): I am just wondering how subthemes 2.1 and 2.2 fit within the theme of being rejected by external world? Maybe it is just the phrasing, but it is not clear for me.

With such a broad range of participants it would be helpful to know a bit more about participants when providing direct quotes, age in particular, when talking about friendships etc.

Page 33-34 of the discussion reads like a list of the themes and doesn’t add much to the discussion.

I hope my feedback is helpful in enhancing the reporting of this very interesting review

6. PLOS authors have the option to publish the peer review history of their article (what does this mean?). If published, this will include your full peer review and any attached files.

Reviewer #1: **Yes: **Keith Gaynor

Reviewer #2: No

---

## [Author Response · Author response to Decision Letter 0]

27 Nov 2024

Please refer to the file labeled 'Response to Reviewers'.

---

## [Editor Report · Decision Letter 1]

2 Dec 2024

The experience of loneliness among people with psychosis: qualitative meta-synthesis

PONE-D-24-33999R1

Dear Dr. Ahmed,

We’re pleased to inform you that your manuscript has been judged scientifically suitable for publication and will be formally accepted for publication once it meets all outstanding technical requirements.

Kind regards,

Vittorio Lenzo

Academic Editor

PLOS ONE
---

## [Editor Report · Acceptance letter]

10 Dec 2024

PONE-D-24-33999R1 

PLOS ONE

Dear Dr. Ahmed, 

I'm pleased to inform you that your manuscript has been deemed suitable for publication in PLOS ONE. Congratulations! Your manuscript is now being handed over to our production team.

Kind regards, 

on behalf of

Professor Vittorio Lenzo 

Academic Editor

PLOS ONE